# A comparison of new cardiovascular endurance test using the 2-minute marching test vs. 6-minute walk test in healthy volunteers: A crossover randomized controlled trial

Suchai Surapichpong[1☯]*, Sucheela Jisarojito[1☯], Chaiyanut Surapichpong[2☯]

1 Rehabilitation Center, Bangkok Hospital Headquarters, Bangkok Dusit Medical Services, Bangkok, Thailand, 2 Department of Outpatient, Bangbo Hospital, Samutprakarn, Thailand

☯ These authors contributed equally to this work.
* s.suchai.s@hotmail.com

**Data Availability Statement:** The data are available at Dryad. DOI: 10.5061/dryad.31zcrjdv2.

## Abstract

### Trial design

This was a 2×2 randomized crossover control trial.

### Objective

To compare the cardiovascular endurance of healthy volunteers using a 2-minute marching test (2MMT) and a 6-minute walk test (6MWT).

### Methods

This study included 254 participants of both sexes, aged 20–50 years, with a height and body mass index (BMI) of $\geq$150 cm and $\leq$25 kg/m$^2$, respectively. Participants were hospital staff who could perform activities independently and had normal annual chest radiographs and electrocardiograms. A group-randomized design was used to assign participants to Sequence 1 (AB) or 2 (BA). The tests were conducted over 2 consecutive days, with a 1-day washout period. On day 1, the participants randomly underwent either a 6MWT or 2MMT in a single-anonymized setup, and on day 2, the tests were performed in reverse order. We analyzed maximal oxygen consumption (VO$_{2max}$) as the primary outcome and heart rate (HR), respiratory rate (RR), blood pressure (BP), oxygen saturation, dyspnea, and leg fatigue as secondary outcomes.

### Results

Data were collected from 127 participants, categorized into two groups for different testing sequences. The first (AB) and second groups had 63 and 64 participants, respectively. The estimated VO$_{2max}$ was equivalent between both groups. The 2MMT and 6MWT estimated VO$_{2max}$ with a mean of 41.00 ± 3.95 mL/kg/min and 40.65 ± 3.98 mL/kg/min, respectively.

**Funding:** The author(s) received no specific funding for this work.

**Competing interests:** the authors have declared that no competing interests exist.

The mean difference was -0.35 mL/kg/min (95% confidence interval: -1.09 to 0.38; p <0.001), and no treatment and carryover effect were observed. No significant changes were observed in HR, RR, and systolic BP (p = 0.295, p = 0.361 and p = 0.389, respectively). However, significant changes were found in the ratings of perceived exertion (p <0.001) and leg fatigue scale (p <0.001).

## Conclusion

The 2MMT is practical, simple, and equivalent to the 6MWT in estimating $VO_{2max}$.

## Trial registration

TCTR20220528004 https://www.thaiclinicaltrials.org.

## Introduction

The cardiovascular endurance test evaluates the maximal oxygen consumption ($VO_{2max}$) among individuals engaging in continuous activities or exercises, such as walking, going up and down stairs, running, and aerobic exercise, which require the cardiovascular endurance system to transport oxygen to the muscles during exercise [1]. The maximum amount of oxygen that a person can consume does not change despite increased workload over time. $VO_{2max}$ (expressed as ml/kg/min) is an indicator of cardiovascular endurance [2] and can be estimated through maximal or submaximal tests using direct or indirect methods [3].

$VO_{2max}$ is the main evaluation parameter in the cardiovascular endurance test. It can be assessed in a laboratory, and a treadmill or bicycle ergometer test is required to determine the peak oxygen uptake. For the test, the heart rate (HR) is determined at 80–90% of the maximum HR and tested with progressive intensity until the maximum HR level is achieved or until an adverse event occurs [4]. In addition to laboratory testing, other highly accurate tests are primarily used in clinical practice, such as the step test, 6-minute walk test (6MWT), and 2-minute step test (2MST) [5].

The walk test can also be used to assess submaximal cardiorespiratory or endurance fitness. This test was initially developed to assess patients with lung disease [6]. The original 12-minute walk test (12MWT) was further developed by reducing the test time to 6 min. The 6MWT appears to be a more appropriate instrument for assessing the exercise response in patients with lung disease than the 12MWT [7]. Walking is a more sensitive exercise modality for evaluating the response to bronchodilators in patients with lung disease [7, 8]. The 6MWT measures the distance a patient can quickly walk on a flat, solid surface within 6 min. It evaluates the global and integrated responses of all systems involved during exercise, including the pulmonary and cardiovascular systems, systemic blood circulation, and neuromuscular and muscle metabolism [9]. The 6MWT is also a predictor for assessing prognostic indicators and may be useful in the serial evaluation of patient's status and response to therapeutic interventions [8]. The 6MWT is a standardized test for assessing cardiovascular endurance and is widely used in patients with lung or heart disease, including the aging population. Overall, the 6MWT has a moderate ability to predict maximum oxygen and functional capacity in patients with congestive heart failure who can walk for <490 m [10, 11]. The American Thoracic Society recommends using 6MWT to assess the functional status and predict the morbidity and mortality rates in patients with heart or lung disease, including older adults [11]. In addition to

using 6MWT to assess cardiovascular endurance, the 2MST has also been used to assess cardiovascular endurance in older adults. During the epidemic, the American Physical Therapy Association also recommended using the 2MST to assess cardiovascular endurance in patients with coronavirus disease 2019 (COVID-19) [12, 13]. However, 2MST assessment has limitations regarding the high leg rising setting. A discrepancy in the measurement process may cause error in the assessment results. Previous studies have shown that the 2MST can assess cardiovascular endurance in older adults [13]. However, the reliability of 2MST assessment in healthy adults has not been studied.

Based on these limitations of evaluating cardiovascular endurance using the 6MWT and 2MST to develop a new method for assessing cardiovascular endurance expected that limitations based on testing area, equipment, and process could cause inaccuracies. Therefore, we designed a new cardiovascular endurance test, the 2MMT. Participants were instructed to elevate their legs with their toes 30 cm from the ground (based on the stair height of the Young Men's Christian Association (YMCA) step test) [14], and continuously march in place as soon as possible for 2 min [9]. Therefore, we aimed to perform an equivalence test of the cardiovascular endurance of healthy volunteers using the 2MMT and 6MWT.

## Materials and methods

The trial protocol and supporting Consolidated Standards of Reporting Trials (CONSORT) checklist are available as supporting information (S1 File CONSORT Checklist) and the CONSORT diagram is illustrated in Fig 1.

### Trial design

We conducted a group of randomized controlled trials, including a crossover study for the equivalence test of cardiovascular endurance ($VO_{2max}$) between the 6MWT and 2MM, with a 1-day washout period between the 6MWT and 2MMT. The trial design is outlined in Fig 2.

### Overview of study design

This study was conducted during two visits, 1 day apart. Notably, 254 healthy volunteers were enrolled from the Bangkok Hospital Headquarters. They were recruited between April 2022 and July 2022 through an announcement within hospital. All eligible participants were randomly assigned to one of two sequences using group randomization in a single anonymized setup. Sequences 1 (AB) and 2 (BA) involved the 6MWT and 2MMT, respectively. On day 1, the sequences were performed randomly, with a 1-day washout period between the tests. On day 2, the tests were performed in reverse order. The primary outcome, $VO_{2max}$, was recorded in the posttest, and the secondary outcomes, including heart rate (HR), respiratory rate (RR), oxygen saturation ($SpO_2$), systolic blood pressure (SBP), diastolic blood pressure (DBP), rating of perceived exertion (RPE), and leg fatigue scale (LFS), were recorded in 1 min, 5 min, and 10 min for pretest and posttest. The study protocol is outlined in Fig 3.

### Ethics statement

This study was conducted following the principles embodied in the Declaration of Helsinki. The study protocol was approved by the Human Ethics Committee/Institutional Review Board of Bangkok Health Research Center (BHQIRB 2022-01-01, COA 2022–10) and was registered at the Thai Clinical Trials Registry (TCTR) (Thaiclinicaltrials.org; trial registration ID: TCTR20220528004). All study participants provided written informed consent.

Enrollment | Assessment for eligibility (n=254)

Excluded (n=127)
- Not meeting inclusion criteria (n= 6)
- Declined to participate (n=0)
- Other reasons (n=121)

Randomized (n=127)

Allocation

Allocated to sequence 1 (AB) (n= 64)

Allocated to sequence 2 (BA) (n= 64)

Period 1; 6MWT (A)

Period 1; 2MMT (B)

Wash-out period

Period 2; 6MWT (A)

Period 2; 2MMT (B)

Analysis

Analyzed (n=127)

- Excluded from analysis (give reason) (n=0)

**Fig 1. CONSORT diagram of the study.** 6MWT = 6-minute walk test, 2MMT = 2-minute marching test, CONSORT = Consolidated Standards of Reporting Trials https://doi.org/10.5061/dryad.31zcrjdv2.

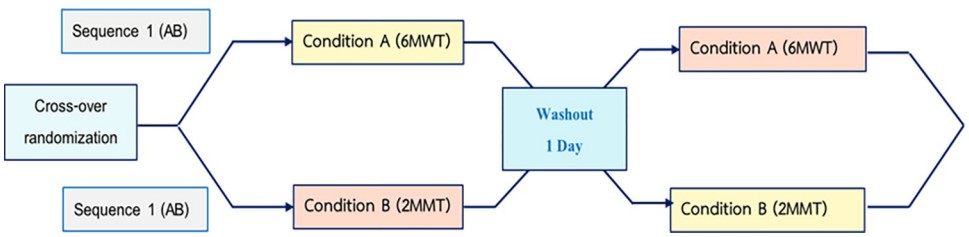

**Fig 2. The trial design.** 6MWT = 6-minute walk test, 2MMT = 2-minute marching test https://doi.org/10.5061/dryad.31zcrjdv2.

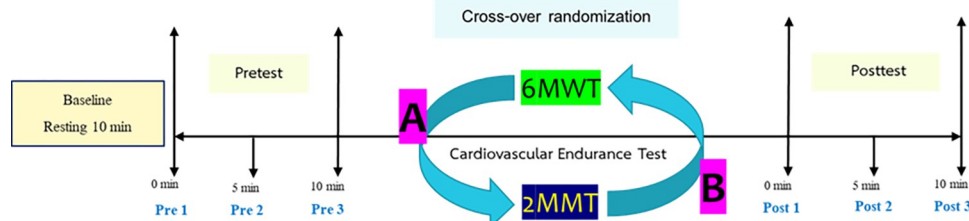

**Fig 3. The study protocol.** 6MWT = 6-minute walk test, 2MMT = 2-minute marching test https://doi.org/10.5061/dryad.31zcrjdv2.

## Sample size

The sample size required for the equivalence study was estimated using nQuery software and calculated using two one-sided equivalence tests for crossover design. To calculate the sample size, we set the alpha error probability, statistical power, the lower equivalence limit, and upper equivalence limit at 5%,90%, -2.00, and +2.00, respectively, using the clinical margin (minimal clinically important difference [MCID] of VO2max from a previous study, which was 2 ml/kg/min [15], and standard deviation was 8.6 [16]. Based on these values, we needed 101 participants for the crossover design, allowing for a 20% dropout rate. Therefore, we decided to randomize 127 patients per arm, resulting in 254 participants. However, due to the COVID-19 pandemic, data collection was incomplete, and we could only analyze 127 data sets in this study.

## Inclusion and exclusion criteria

The inclusion criteria were male and female healthy volunteers, aged 20–50 years, with height: ≥150 cm and, BMI ≤25 kg/m2. Participants were hospital staff who could perform activities independently and had normal annual chest radiographs and electrocardiograms. The exclusion criteria were significantly unstable vital signs, a history of COVID-19, and underlying heart disease or neuromuscular/skeletal impairment.

## Procedure and measurement

We conducted a 2MMT and compared the results with those of the standard test, the 6MWT, to test the equivalence of both tests in estimating VO2max

## Condition A

According to the standard protocol, the 6MWT was performed indoors on a flat surface in a 30-m straight corridor, with 180˚ turns every 30 m. [10]. The walk test was performed with stable vital signs, and SpO2 was maintained at >95%, all monitored by a cardiopulmonary physical therapist.

## Condition B

The 2MMT was developed to determine the number of steps performed within 2 min. After the "start" command, the participants began marching in place and lifting their knees to an appropriate height of 30 cm. The participants were instructed to perform as many steps as possible (reaching a height of 30 cm) within 2 min. The participants were allowed to perform a few training steps to adjust to the marching technique and verify their ability to complete the task. The participants marched at their own pace; they could slow down or even stop, if

necessary, and continue marching until the end of the 2-min test period. The investigator determined the number of steps performed, informed the participants about the time left until the end of the trial, and motivated them to achieve the best possible result. The test results were expressed as the number of performed steps during which the right foot touched the ground.

When the participants exhibited severe symptoms of exercise intolerance in both tests, such as severe dyspnea, fatigue, or other alarming symptoms, they were allowed to slow down or stop and rest. However, they were encouraged to resume the test as soon as possible. Adverse events were monitored during and after test completion. Both tests were terminated and interpreted as incomplete if any of the following symptoms were present: chest pain, intolerable dyspnea, leg cramps, staggering, diaphoresis, and ashen appearance.

Data regarding the sex, age, BMI, HR, and RR were collected, and SpO2 was assessed using the NONIN Onyx2 9590 Oximeter, SBP and DBP were measured using the Philip Patient Monitor Efficia CM100, RPE, and LFS were assessed using the Borg's scale. All parameters were recorded at 1 min, 5 min, and 10 min for pretest and posttest.

VO2max estimated the cardiovascular endurance using the following formula:

- $VO_{2max}$ estimated in the 6MWT: 70.161 + (0.023 × 6MWT [m])—(0.276 × weight [kg])—(6.79 × sex, where m = 0, f = 1)—(0.193 × resting HR [beats per minute]—(0.191 × age [years]) [15]. where resting HR is the 10-min resting HR of posttest.

- $VO_{2max}$ estimated in the 2MMT: 13.341 + 0.138 × total up and down steps (UDS)–(0.183 × BMI) [16].

## Data analysis

Due to the COVID-19 pandemic in Thailand and hospital policies, only 127 of the 254 participants, who were healthy volunteers, could complete data collection. Descriptive statistics were used to evaluate demographic characteristics. Continuous variables were reported as mean ± standard deviation, whereas binary variables were reported as percentages. The primary outcome ($VO_{2max}$), evaluated using Statgraphics software, was analyzed through a two-one-sided t-test procedure. The analysis was conducted with an equivalence bound of ± 2 mL/kg/min from the margin of $VO_{2max}$ observed in a previous study [17]. The carryover and treatment effects were insignificant, and the equivalence result was significant for the test. For the secondary outcome, all parameters were analyzed using a linear mixed-effect model to compare 2MMT and 6MWT with STATA software.

## Patient and public involvement

Co-investigators, research assistants, and five physiotherapists in the team, participated in developing the 2MMT, an assessment protocol and created a guide of commands for evaluation. However, participants suggested revising the guide of commands in assessing cardiovascular endurance to enable, easy understanding and correct compliance.

## Results

### Characteristics

Notably,127 healthy volunteers completed the data collection and protocol without experiencing adverse events. The study involved 31 males and 96 females aged 20–49 years, with a mean age of 29.65 ± 5.85 years. The participants' average, weight, height, and BMI were, 56.96 ± 8.02 kg, 163.92 ± 6.72 cm, and 21.05 ± 2.08 kg/m$^2$, respectively. Participants had an average of

**Table 1. Baseline characteristics (n = 127).**

| Parameter | Sequence 1 (n = 64) Mean (SD) | Sequence 2 (n = 63) Mean (SD) | Total (n = 127) Mean (SD) |
|---|---|---|---|
| Sex (M/F) | 19/44 | 12/52 | 31/96 |
| Age (Years) | 29.31 (5.44) | 30.00 (6.26) | 29.65 (5.85) |
| Bogy weight (kgs) | 55.90 (5.78) | 58.04 (8.37) | 56.96 (8.02) |
| Height (cm) | 163.58 (6.36) | 164.28 (7.01) | 163.92 (6.72) |
| BMI (kg/m$^2$) | 20.73 (2.09) | 21.39 (2.03) | 21.05 (2.08) |
| 2MMT (steps) | 226.01 (21.18) | 240.42 (59.71) | 233.16 (46.94) |
| 6MWT (distance) | 568.69 (56.22) | 562.23 (59.78) | 565.48 (57.88) |

Abbreviation: BMI, body mass index; SD, standard deviation; 2MMT, 2-minute marching test; 6MWT, 6-minute walk test

233.16 ± 46.94 steps, for the 2MMT and, an average distance 565.48 ± 56.88 m. for 6MWT. The participants' characteristics are presented in Table 1.

## Core intervention finding

Overall, 127 participants were included in this study; 64 participants underwent sequence 1 (AB) (6MWT followed by 2MMT), whereas 63 underwent sequence 2 (BA) (2MMT followed by 6MWT). The results are presented in Table 2.

This study found that the 2MMT and 6MWT were equivalence in estimating VO$_{2max}$ The mean VO2max for the 2MMT was 41.00± 3.95 mL/kg/min, whereas it was 40.65 ±3.98 mL/kg/min for the 6MWT. The mean difference between the two tests was only 0.35 mL/kg/min, with a 95% confidence interval [CI] of -1.09 to 0.38 mL/kg/min. The p-value was < 0.001, indicating no treatment or carryover effect.

For sequence 1 (AB), the mean VO$_{2max}$ for both tests were 41.09 ± 4.13 mL/kg/min and 41.06 ± 3.88 mL/kg/min, respectively, with a mean difference of only 0.27 mL/kg/min (95% CI: 1.36, 1.42). The statistical analysis showed no significant difference between both methods (p-value 0.96), indicating that both tests yielded similar results for sequence 1 (AB).

Similarly, for sequence 2 (BA), the mean VO$_{2max}$ was 40.87 ± 3.80 mL/kg/min and 41.29 ± 3.94 mL/kg/min for the 2MMT and 6MWT, respectively, with a mean difference of 0.58 mL/kg/min (95% CI: -0.50, 1.65). The statistical analysis was not significant (p-value = 0.28), indicating that both tests yielded similar results for sequence 2 (BA).

**Table 2. Equivalence test of VO$_{2max}$ between 2MMT and 6MWT.**

| VO$_{2max}$ | 2MMT | 6MWT | Difference between groups (95% CI) | Treatment effect (p-value) | Carryover effect (p-value) | Equivalence (p-value) |
|---|---|---|---|---|---|---|
| | Mean (SD) | Mean (SD) | | | | |
| Total (n = 127) | 41.00 (3.95) | 40.65 (3.98) | -0.35(-1.09 to 0.38)[a] | 0.42 | 0.37 | <0.001* |
| Sequence 1 (AB) (n = 64) | 41.09 (4.13) | 41.06 (3.88) | 0.27 (-1.36 to 1.42) [b] | 0.96 | - | - |
| Sequence 2 (BA) (n = 63) | 40.87 (3.80) | 41.29 (3.94) | 0.58 (-0.50 to 1.65) [b] | 0.28 | - | - |

Abbreviations: 2MMT, 2-minute marching test; 6MWT, 6-minute walk test; VO$_{2max}$, maximal oxygen consumption; SD, standard deviation; CI, confidence interval.

[a] Analyses were conducted using the two-one-sided t-test procedure (TOST).

[b] Analyses were conducted using a paired t-test.

* Statistical significance set at p< 0.05

https://doi.org/10.5061/dryad.31zcrjdv2

**Table 3. Mean and standard deviation of 6MWT and 2MMT.**

| Outcome/Time | 2MMT | 6MWT |
|---|---|---|
| | Mean (SD) | Mean (SD) |
| **HR (bpm)** | | |
| 0 min | 111.10 (21.82) | 105.81 (19.66) |
| 5 min | 84.99 (12.75) | 87.25 (11.72) |
| 10 min | 82.73 (11.31) | 84.64 (10.65) |
| **RR (bpm)** | | |
| 0 min | 21.57 (2.64) | 21.43 (2.58) |
| 5 min | 18.48 (1.79) | 18.22 (2.09) |
| 10 min | 17.35 (1.50) | 17.19 (1.76) |
| **SpO$_2$ (%)** | | |
| 0 min | 99.35 (0.97) | 99.57 (0.72) |
| 5 min | 99.34 (0.90) | 99.44 (0.78) |
| 10 min | 99.41 (0.84) | 99.31 (0.80) |
| **SBP (mmHg)** | | |
| 0 min | 132.86 (16.87) | 135.06 (16.48) |
| 5 min | 117.50 (12.65) | 117.20 (12.49) |
| 10 min | 114.95 (11.62) | 114.43 (10.48) |
| **DBP (mmHg)** | | |
| 0 min | 70.85 (10.26) | 75.27 (11.17) |
| 5 min | 70.57 (9.03) | 71.39 (9.15) |
| 10 min | 70.28 (8.50) | 71.36 (9.57) |
| **RPE** | | |
| 0 min | 1.96 (1.25) | 2.35 (1.40) |
| 5 min | 0.29 (0.51) | 0.56 (0.74) |
| 10 min | 0.04 (0.16) | 0.11 (0.34) |
| **LFS (Borg scale)** | | |
| 0 min | 1.85 (1.23) | 2.11 (1.44) |
| 5 min | 0.42 (0.63) | 0.71 (0.83) |
| 10 min | 0.11 (0.31) | 0.22 (0.53) |

Abbreviations: 2MMT, 2-minute marching test; 6MWT, 6-minute walk test; HR, heart rate; RR, respiratory rate; SpO$_2$, oxygen saturation; SBP, systolic blood pressure; DBP, diastolic blood pressure; LFS, leg fatigue scale; SD, standard deviation.

Regarding the secondary outcomes, we compared the vital sign variables (HR, RR, SpO$_2$, DBP, and SBP), dyspnea scale (RPE), and LFS which were analyzed using a linear mixed-effect model adjusted for baseline value. The results are outlined in Tables 3 and 4.

**HR.** The comparison results in the posttest at 0 min of 2MMT and 6MWT presented a mean HR of 111.10 ± 21.82 bpm, and 105.81 ± 19.66 bpm, respectively. The 5 min posttest had a mean HR of 84.99 ± 12.75 bpm and 87.25 ± 11.72 bpm in 2MMT and 6MWT, respectively, and that for 10 min posttest was 82.73 ± 11.31 bpm and 84.64 ± 10.65 bpm, respectively. Changes in HR during the 2MMT had a statistically significant mean reduction of -2.672 (95% CI: -2.965, -2.379) (p-value <0.001), and those in 6MWT had a mean reduction of -2.282. (95%CI: -2.575, -1.989), which was statistically significant (p-value <0.001). The HR between the 2MMT and 6MWT had, a mean difference of -0.849 (95%CI: -2.438, 0.740) and was, not statistically significant (p-value = 0.295).

**RR.** The comparison results in the posttest at 0 min of the 2MMT and 6MWT presented a mean of RR 21.57 ± 2.64 bpm and 21.43 ± 2.58 bpm, respectively. The mean RR during the 5

**Table 4. Comparison of secondary outcomes between 6MWT and 2MMT.**

| Outcome | 2MMT | | 6MWT | | Difference between groups (95%CI) [b] | p-value [b] |
|---|---|---|---|---|---|---|
| | Mean change (95% CI) [a] | p-value [a] | Mean change (95% CI) [a] | p-value [a] | | |
| HR (bpm) | -2.672 (-2.965 to -2.379) | <0.001* | -2.282 (-2.575 to -1.989) | <0.001* | -0.849 (-2.438 to 0.740) | 0.295 |
| RR (bpm) | -0.415 (-0.454 to -0.376) | <0.001* | -0.431 (-0.469 to -0.392) | <0.001* | 0.121 (-0.138 to 0.379) | 0.361 |
| SpO₂ (%) | -0.010 (-0.024 to 0.004) | 0.163 | -0.011 (-0.024 to 0.003) | 0.141 | -0.106 (-0.195 to -0.017) | 0.020* |
| SBP (mmHg) | -1.838 (-2.077 to -1.600) | <0.001* | -2.014 (-2.253 to -1.775) | <0.001* | 0.560 (-0.715 to 1.834) | 0.389 |
| DBP (mmHg) | -0.263 (-0.412 to -0.114) | 0.001* | -0.185 (-0.334 to -0.036) | 0.015* | -1.759 (-2.589 to -0.928) | <0.001* |
| RPE | -0.215 (-0.237 to -0.194) | <0.001* | -0.201 (-0.222 to -0.180) | <0.001* | -0.227 (-0.350 to -0.105) | <0.001* |
| LFS (Borg scale) | -0.193 (-0.216 to -0.169) | <0.001* | -0.171 (-0.195 to -0.148) | <0.001* | -0.226 (-0.341 to -0.111) | <0.001* |

Abbreviations: 2MMT, 2-minute marching test; 6MWT, 6-minute walk test; HR, heart rate; RR, respiratory rate; SpO₂, oxygen saturation; SBP, systolic blood pressure; DBP, diastolic blood pressure; LFS, leg fatigue scale; CI, confidence interval.

[a] Analyses were conducted using a paired t-test.

[b] Analyses were conducted using a linear mixed-effects model adjusted for baseline value.

* Statistical significance set at $p < 0.05$.

min posttest was 18.48 ± 1.79 bpm and 18.22 ± 2.09 bpm in 2MMT and 6MWT, respectively, and that for the 10 min posttest was 17.35 ± 1.50 bpm and 17.19 ± 1.76 bpm, respectively. RR changes during the 2MMT had a statistically significant mean reduction of -0.415 (95%CI: -0.454, -0.376) (p-value <0.001), and those in 6MWT had a mean decrease of -0.431 (95%CI: -0.469, -0.392) which was statistically significant (p-value <0.001). The RR change between the 2MMT and 6MWT had, a mean difference of 0.121 (95%CI: -0.138, 0.379) and was, not statistically significant (p-value = 0.361).

**SpO₂.** The comparison results in the posttest at 0 min of the 2MMT and 6MWT presented a mean of SpO₂ of 99.35 ± 0.97% and 99.57 ± 0.72%, respectively. The mean SpO₂ at 5 min posttest was 99.34 ± 0.90% and 99.44 ± 0.78% in 2MMT and 6MWT, respectively, and that for the 10 min in posttest was 99.41 ± 0.84% and 99.31 ± 0.80%. Changes in SpO₂ 2MMT had statistically non- significant mean reduction of -0.010 (95%CI: -0.024, 0.004) (p-value = 0.163), and those in the 6MWT had a non-significant mean decrease of -0.011 (95%CI: -0.024, 0.003) (p-value = 0.141) the change in SpO₂ between the 2MMT and 6MWT, a mean difference of -0.106 (95%CI: -0.195, -0.017) was statistically significant (p-value = 0.020).

**SBP.** The comparison results in the posttest at 0 min of the 2MMT and 6MWT presented a mean SBP 132.86 ± 16.87 mmHg and 135.06 ± 16.48 mmHg, respectively. The mean SBP at the 5 min posttest 117.50 ± 12.65 mmHg and 117.20 ± 12.49 mmHg in 2MMT and 6MWT, respectively, and that for 10 min posttest was 114.95 ± 11.62 mmHg and 114.43 ± 10.48 mmHg, respectively. SBP changed during 2MMT had a statistically significant mean reduction of -1.838 (95%CI: -2.077, -1.600) (p-value <0.001), and those during 6MWT had a mean decrease of -2.014 (95%CI: -2.253, -1.775), which was statistically significant (p-value <0.001). The SBP changes between the 2MMT and 6MWT had, a mean difference of 0.560 (95%CI: -0.715, 1.834) and was, not statistically significant (p-value = 0.389).

**DBP.** The comparison results in the at 0 min of 2MMT and 6MWT presented a mean DBP of 70.85 ± 10.26 mmHg and 75.27 ± 11.17 mmHg, respectively. The mean DBP at the 5 min posttest was 70.57 ± 9.03 mmHg and 71.39 ± 9.15 mmHg in 2MMT and 6MWT, respectively, and that for the 10 min posttest was 70.28 ± 8.50 mmHg and 71.36 ± 9.57 mmHg, respectively. DBP changes during 2MMT had a statistically significant mean reduction of 0.263 (95%CI: -0.412, -0.114) (p-value = 0.001). The DBP changes in 6MWT had a mean reduction of 0.185 (95%CI: -0.334, -0.036), which was statistically significant (p-value = 0.015).

The DBP changes between the 2MMT and 6MWT had a statistically significant mean difference of -1.759 (95%CI: -2.589, -0.928) (p-value < 0.001).

The dyspnea scale was presented using RPE and the comparison result in the posttest at 0 min of 2MMT and 6MWT presented a mean RPE of 1.96 ± 1.25 and 2.35 ± 1.40, respectively. The mean RPE at the 5 min posttest was 0.29 ± 0.51 and 0.56 ± 0.74 in 2MMT and 6MWT, respectively, and that for the 10 min posttest was 0.04 ± 0.16 and 0.11 ± 0.34 in 2MMT and 6MWT, respectively. RPE changes during 2MMT had a statistically significant mean reduction of 0.215 (95%CI: -0.237, -0.194) (p-value <0.001), those during 6MWT had a statistically significant mean decrease of -0.201 (95%CI: -0.222, -0.180) (p-value < 0.001), and those between the 2MMT and 6MWT had, a statistically significant mean difference of -0.227 (95%CI: -0.350, -0.105) (p-value < 0.001).

An LFS was modified from Borg's scale, and the comparison results in the posttest at 0 min of 2MMT and 6MWT presented a mean LFS of 1.85 ± 1.23 and 2.11 ± 1.44, respectively. The 5 min posttest presented a mean of 0.42 ± 0.63 and 0.71 ± 0.83 in 2MMT and 6MWT, respectively, while the 10 min posttest had 0.11 ± 0.31 and 0.22 ± 0.53 in 2MMT and 6MWT, respectively. LFS changes during 2MMT had a statistically significant mean decrease of 0.193 (95% CI: -0.216, -0.169) (p-value <0.001), those during 6MWT had a statistically significant mean decrease of -0.171 (95%CI: -0.195, -0.148) (p-value < 0.001), and those between the 2MMT and 6MWT had a statistically significant, mean difference of -0.226 (95%CI: -0.341, -0.111) (p-value < 0.001).

## Discussion

### Primary outcome: Main findings for estimating $VO_{2max}$

The cardiovascular endurance test is frequently used to predict health status, mortality, and decrease prevalence or incidence. It is a safe, convenient, and valid measurement tool in epidemiological studies [18, 19].

In this study, we primary aimed to evaluate the cardiovascular endurance determined with ($VO_{2max}$) in healthy volunteers using the 6MWT and 2MMT. Our main results confirmed that the 2MMT was equivalent to the 6MWT in estimating $VO_{2max}$. Furthermore, our results provide an opportunity for discussion regarding whether the step test protocol has developed over time, as some previously developed step tests have adjusted the step height according to sex, stature, and step rate based on age [20, 21]. In this study, we developed a step test and fixed the height at 30 cm. We also instructed the participants to lift both legs alternatively up and down as fast as possible for 2 min to estimate $VO_{2max}$, which was performed as an equivalence test using the standard test for estimating oxygen (6MWT).

We estimated $VO_{2max}$ using the following formula for 6MWT and 2MMT, respectively:

- 6MWT: 70.161 + (0.023 × 6MWT [m])—(0.276 × weight [kg])—(6.79 × sex, where m = 0, f = 1)—(0.193 × resting HR [beats per minute])—(0.191 × age [years]) [15].

- 2MMT: 13.341 + 0.138 × total UDS–(0.183 × BMI) [16].

Both formulas for estimating $VO_{2max}$ have been verified for validity and have been found to accurately estimate $VO_{2max}$ accurately.

The results of this study indicated that the 6MWT and 2MMT were equivalent in estimating VO2max in healthy volunteers. The 2MMT requires a shorter time and is easier to conduct, however, it was equivalent to the 6MWT in estimating $VO_{2max}$ in healthy volunteers. No sequence effects were observed in this study. We used random assignments because each test had a sufficient washout period. Therefore, the 2MMT could be used to evaluate

cardiovascular endurance and estimate $VO_{2max}$ in aging populations, patients with lung or heart disease, or other groups of patients who cannot be assessed using standard methods. In a previous study, Bohannon et al. compared the 6MWT and YMCA step test and found that most older adults with an average age of 70.4 ± 14.2 years did not complete the YMCA step test [22]. This is also consistent with the findings of Beutner et al., who reported that 17% of participants (particularly older adults and those with a high BMI) did not complete the 3-min YMCA test [21]. Therefore, the 2MMT may be used to assess cardiovascular endurance in these groups of patients.

**Secondary outcomes.** The secondary outcomes can be summarized using comparison results in posttest at 0, 5, and 10 min between 2MMT and 6MWT as a vital sign (HR, RR, $SpO_2$, SBP, and DBP), The dyspnea scale was presented using RPE, and the LFS was modified through Borg's scale. The results showed a statistically no significant between change of HR, RR, and SBP in 2MMT and 6MWT, and between the $SpO_2$, DBP, RPE and LFS changes in 2MMT and 6MWT. The results can be described as follows:

**HR.** HR was significant in both test between the pretest and posttest. Regarding the hemodynamic response to the effort in each test, our results indicated that, although both tests generate physical exertion, increased HR was more significant at the end of the 2MMT and 6MWT, consistent with what has been observed in other populations [23]. This result is consistent with that of a previous study, which showed that HR increased significantly after light to moderate intensity exercise, HR also increases with other factors, including body position, certain physical condition, health stage, and environment [24].

**RR.** RR was significant in both tests between the pretest and posttest. This attribute of ventilation includes an increased work rate during submaximal exercise intensity due to increases in tidal volume and RR [24]. This result is consistent with that of a previous study, which studied reference values for the primary variables measured in the 6MWT in healthy participants. The RR showed greater tachypnea after a 6MWT [17]. However, the RR is a rarely documented parameter in the literature and a possible outcome parameter in the step and walk tests [25].

**$SpO_2$.** $SpO_2$ was not significant in both tests between the pretest and posttest. The 6MWT remains the gold standard for titrating oxygen during exercise [26]. Our study showed no significant difference in $SpO_2$ at the end of the 6MWT compared with the end of the 2MMT; thus, comparing oxygen desaturation during the 6MWT and 2MMT with similar durations could be interesting. Studies on other field tests, such as the 6MST or 3-min step test, are following our results regarding the capacity of these tests to measure oxygen desaturation during exercise, showed results comparable to our [27].

**BP.** DBP showed a significant difference, but SBP did not. The increase in SBP after a relative increase in HR affected both tests due to physiological effects; similar results were obtained by Jothi et al., who conducted a study on 15 male kabaddi players aged 20–25 years and suggested the same finding that SBP increases with increasing dynamic work due to increased cardiac output. In contrast, the diastolic pressure usually remains approximately the same or maybe zero in some normal participants [28].

**Dyspnea and LFS.** In this study, the dyspnea scale used was RPE scale, and the LFS used a modified Borg's scale. We found a significant difference in the mean dyspnea scale score and LFS between both groups. The nature of the test's characteristics and duration creates individual differences in perception. For the 6MWT, the participants were required to walk a special length of the corridor of 30 m., within 6 min. The 2MMT was performed, and a difference in the dyspnea perception course of the test characteristics was observed within 2 min. This may be because the characteristics of both tests and, those that use the lower leg muscles that require endurance and strength, are similar [29].

### Safety

No adverse events occurred in this study. This may be because the participants included healthy volunteers with no risk of heart or lung complications. However, the new cardiovascular endurance test using the 2MMT will likely reduce the adverse events previously reported in patients with chronic obstructive pulmonary disease [30]. It is also likely to evaluate cardiorespiratory endurance in older adults or those at risk of osteoarthritis.

### Implications for practice

The results of this study indicated that the 2MMT was equivalent to the 6MWT in estimating $VO_{2max}$. No difference was observed in cardiovascular responses in either test. Therefore, the 2MMT can be used to evaluate functional ability or cardiovascular endurance in aging patients, those with lung complications, and those at risk for heart disease or hypertension. It could be a time-saving test for follow-up assessment during rehabilitation, home isolation, or routine clinical assessment.

### Public implications

According to research, both testing methods can estimate the $VO_{2max}$. Medical professionals can also use the 2MMT test in patients who cannot undergo the standard testing procedure. This test is also useful for evaluating the respiratory and vascular systems and can help promote health awareness and self-care in society.

### Study limitations

Our study had some limitations. This study investigated only healthy volunteers and was a preliminary study; however, the results indicated that 2MMT was equivalent to 6MWT in evaluating cardiovascular endurance. It remains unclear whether 2MMT can effectively evaluate patients. Therefore, further studies are required to investigate the sensitivity and specificity of 2MMT and determine the MCID for this test in clinical practice, including in postoperative patients and those with lung or heart disease.

## Conclusions

We investigated the equivalence of the new cardiovascular endurance test using the 2MMT and 6MWT in healthy volunteers. These tests can provide valid estimates of $VO_{2max}$ in epidemiological studies. The 2MMT may be more relevant than expected. However, further studies are needed to determine the 2MMT's sensitivity and MCID.

## Supporting information

**S1 File. CONSORT checklist.**
(PDF)

**S2 File.**
(PDF)

## Acknowledgments

This research was successfully conducted with the kind support of Dr. Matinee Maipang, Hospital Deputy Chief Executive Officer (CEO) Group 1 and Hospital Director, Bangkok Hospital Headquarters, who permitted the research team to use the hospital's facilities to conduct the

study. We also express our gratitude to Mr. Suthipol Udompunturak for contributing to the methodology and statistical analysis and for generously and willingly providing his time. We would like to thank Editage (www.editage.com) for English language editing. Lastly, the authors thank the research team for the great collaboration and all participants who agreed to participate in this study.

## Author Contributions

**Conceptualization:** Chaiyanut Surapichpong.

**Formal analysis:** Chaiyanut Surapichpong.

**Methodology:** Suchai Surapichpong, Sucheela Jisarojito.

**Project administration:** Sucheela Jisarojito.

**Validation:** Sucheela Jisarojito.

**Writing – original draft:** Suchai Surapichpong.

**Writing – review & editing:** Suchai Surapichpong, Chaiyanut Surapichpong.

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
