## [Decision Letter · Decision Letter 0]

25 Mar 2024

PONE-D-23-24966A comparison of new cardiovascular endurance test using the 2-minute marching test vs. 6-minute walk test in healthy volunteers: A crossover randomized controlled trialPLOS ONE

Dear Dr. Surapichpong¹¶,

Thank you for submitting your manuscript to PLOS ONE. After careful consideration, we feel that it has merit but does not fully meet PLOS ONE’s publication criteria as it currently stands. Therefore, we invite you to submit a revised version of the manuscript that addresses the points raised during the review process.

We look forward to receiving your revised manuscript.

Kind regards,

Tamer I. Abo Elyazed, Ph.d

Guest Editor

PLOS ONE

Journal Requirements:

Additional Editor Comments (if provided):

The results should be revised by statistician

Reviewers' comments:

Reviewer's Responses to Questions

**Comments to the Author**

1. Is the manuscript technically sound, and do the data support the conclusions?

Reviewer #1: No

Reviewer #2: Yes

Reviewer #3: Yes

2. Has the statistical analysis been performed appropriately and rigorously? 

Reviewer #1: No

Reviewer #2: Yes

Reviewer #3: Yes

3. Have the authors made all data underlying the findings in their manuscript fully available?

Reviewer #1: No

Reviewer #2: Yes

Reviewer #3: Yes

4. Is the manuscript presented in an intelligible fashion and written in standard English?

Reviewer #1: No

Reviewer #2: Yes

Reviewer #3: Yes

5. Review Comments to the Author

Reviewer #1: This is an interesting study looking at the effect of 6 minutes walk versus 2 minute marching. However the design and statistical methods is not appropriately described making it challenging to review.

They are some fundamental methodological information missing, from clarity of the study design to the statistical methods and corresponding analysis.

I would encourage authors to revisit the CONSORT2010 guidelines for reporting cross-over studies.

1. The study design has been labelled at cross-over. with mention of cluster RCT and equivalence. Difficult to assimilate what the clusters were, since participants were recruited from Bangkok hospital headquarters - although the authors notes that each individual was treated as a cluster, I disagree to use the terminology in this way. Cluster could mean a group of individuals in setting e.g. hospital.

2. If this is a cluster randomised trial- the sample size needs to reflect this.

3. Also if this is an equivalence trial the sample size and also hypothesis needs to reflect this.

4. More details are required in the data analysis section, does not clearly justify the use the statistical methods to be used, which is not in lines with the design

5. Also in cross-over design you need to account for carry over effect as well.. although this is waking versus marching, a bit complex to define

6.Interchange use of 127 volunteers and 127 data, also confusing were there is mention of 254 participants when this is a crossover study, so really its still 127 participants.

Reviewer #2: in lines 76,77,78,79 & 80 there is repetitions that is meaningless so it needs editing to clarify your purpose.

In materials : you did not mention the number of males & females enrolled in the study.

In Results : you mentioned the mean age among males and females which spots the importance of sex as a variable.

Moreover in table 1 the number appeared males 36 & females 91 which is of great difference.

In Discussion : line 317 YMCA ...........?

In study limitations : line 386: MCID.......?

In references : no. 2,3,9,26,27 needs to add doi ...........

Reviewer #3: Overall, interesting study entitled A comparison of new cardiovascular endurance test using the 2-minute marching test vs. 6-minute walk test in healthy volunteers: A crossover randomized controlled trial However, the manuscript does not provide a clear overview of their work and requires revision.

Comments to authors:

•Please make sure that the structure for citing published literature in the text, as well as the style of references in the References section, are consistent with the journal's style (see Instructions to Authors).

•English language needs revision for style and syntax.

•Abstract must be rewritten. Add characteristics of the participants (age…..) I suggest focusing the abstract on your study and your results.

•How were the participants randomised?

•

•Please add the originality of the study and add hypothesis at the end of the introduction section. Be please be more specific.

•A substantial revision of the introduction needed.

•Include more characteristics of participants. More information about the participant’s selection needed.

•Table and results are not clear. I would only appreciate to read a detailed statistical approach

•Please specify inclusion/exclusion criteria. The experimental protocol is not clear. The chart flow is not clear.

•Why the tests chosen 6MWTand 2 min marching test? This was not explained? Or measured, the intensity ?.

•Please justify the sample size. There is no calculation of sample size. For the complex statistical analysis, the study is very likely underpowered. What sample size would be adequate?

•Discussion: describing each part of the study

•Please discuss the results of the study in relation to the previous studies.

•Add the public implications of this study.

.

6. PLOS authors have the option to publish the peer review history of their article (what does this mean?). If published, this will include your full peer review and any attached files.

Reviewer #1: No

Reviewer #2: No

Reviewer #3: **Yes: **Salma Abedelmalek

---

## [Author Response · Author response to Decision Letter 0]

5 May 2024

Rebuttal letter 

Response to reviewers

We thank the editor and reviewers for evaluating our manuscript. We have provided line-by-line responses to the comments raised. Please find our responses marked in yellow to every comment/question marked in green. We have copied the review decision from the submission menu of the editorial manager and have used the same letter to write our responses. 

PONE-D-23-24966

A comparison of new cardiovascular endurance test using the 2-minute marching test vs. 6-minute walk test in healthy volunteers: A crossover randomized controlled trial.

PLOS ONE

Dear Dr. Surapichpong¹¶,

Thank you for submitting your manuscript to PLOS ONE. After careful consideration, we feel that it has merit but does not fully meet PLOS ONE’s publication criteria as it currently stands. Therefore, we invite you to submit a revised version of the manuscript that addresses the points raised during the review process.

We look forward to receiving your revised manuscript.

Kind regards,

Tamer I. Abo Elyazed, Ph.d

Guest Editor

PLOS ONE

Journal Requirements:

Responses: We thank you for your valuable suggestions. We have revised the manuscript based on the formatting guidelines of PLOS ONE's style requirements.

Comment: 

If there are ethical or legal restrictions on sharing a de-identified data set, please explain them in detail (e.g., data contain potentially sensitive information, data are owned by a third-party organization, etc.) and who has imposed them (e.g., an ethics committee). Please also provide contact information for a data access committee, ethics committee, or other institutional body to which data requests may be sent. If data are owned by a third party, please indicate how others may request data access

Responses: We thank you for your advice on sharing the “minimal data set” for submission. Accordingly, we have shared the same on dryad.org with the following link: https://datadryad.org/stash/share/8mRXtL-Wh0rc0IO2ywz3t15Uv3jez32rcVDQm7tAx7k. Additionally, a unique digital object identifier (DOI) has been included: doi:10.5061/dryad.31zcrjdv2 

Response: We thank you for your guidance on creating a supporting information file. We have revised our file to adhere to the supporting information guidelines. 

Additional Editor Comments (if provided):

Comment: 

The results should be revised by statistician

Response: We thank you for your valuable suggestion. Accordingly, we have revised the results using the TSOT statistical approach from the statistician.

Reviewers' comments:

Reviewer's Responses to Questions

Comments to the Author

1. Is the manuscript technically sound, and do the data support the conclusions?

Reviewer #1: No

Reviewer #2: Yes

Reviewer #3: Yes

2. Has the statistical analysis been performed appropriately and rigorously? 

Reviewer #1: No

Reviewer #2: Yes

Reviewer #3: Yes

3. Have the authors made all data underlying the findings in their manuscript fully available?

Reviewer #1: No

Reviewer #2: Yes

Reviewer #3: Yes

4. Is the manuscript presented in an intelligible fashion and written in standard English?

Reviewer #1: No

Reviewer #2: Yes

Reviewer #3: Yes

5. Review Comments to the Author

Comment

Reviewer #1: This is an interesting study looking at the effect of 6 minutes walk versus 2 minute marching. However the design and statistical methods is not appropriately described making it challenging to review.

1.They are some fundamental methodological information missing, from clarity of the study design to the statistical methods and corresponding analysis

Response: We thank you for your advice on the study design, methodology, and statistical analysis. We reviewed the study design, methodology, and statistical analysis and revised the statistical analysis approach for the primary outcome, resulting in a change in the results presented in Table 2.

Comment

2.I would encourage authors to revisit the CONSORT2010 guidelines for reporting cross-over studies

Response: We thank you for your advice on revisiting the CONSORT 2010 guidelines for reporting cross-over studies. We have revised the manuscript to adhere to CONSORT 2010 guidelines for randomized crossover design. We have included a CONSORT diagram in Figure 3 and a CONSORT 2010 checklist for randomized crossover design in the S1 File as supporting information.

Comment

3. Also if this is an equivalence trial the sample size and also hypothesis needs to reflect this.

Response: We thank you for your suggestion to review the sample size. We have made revisions to clarify the sample size calculation; we calculated the sample size using two one-sided equivalence tests for crossover design. 

The following change has been made in the manuscript:

To calculate the sample size, we set the alpha error probability, statistical power, lower equivalence limit, and upper equivalence limit at 5%, 90%, -2.00, and +2.00, respectively, using the clinical margin (minimal clinically important difference [MCID]) of VO2max from a previous study, which was 2 mL/kg/min [15], and the standard deviation was 8.6 [16]. Based on these values, we needed 101 participants for the crossover design, allowing for a 20% dropout rate. Therefore, we decided to randomize 127 patients per arm, resulting in 254 participants. 

Comment

4. More details are required in the data analysis section, does not clearly justify the use the statistical methods to be used, which is not in lines with the design

Response: We thank you for your suggestion to review the data analysis section. We have conducted a statistical analysis and used two one-sided equivalence tests for crossover design to test the equivalence of estimating VO2max between 2MMT and 6MWT. We have also provided results for the treatment and carryover effects. Additionally, you can find related results in Table 2.

Comment

5. Also in cross-over design you need to account for carry over effect as well.. although this is waking versus marching, a bit complex to define

Response: We thank you for your suggestion to review the data analysis section. We have conducted a statistical analysis and used two one-sided equivalence tests for crossover design to test the equivalence of estimating VO2max between 2MMT and 6MWT. We have also provided results for the treatment and carryover effects. Additionally, you can find related results in Table 2.

Comment

6. Interchange use of 127 volunteers and 127 data, also confusing were there is mention of 254 participants when this is a crossover study, so really its still 127 participants

Response: We thank you for your comment. We could only collect complete data from 127 healthy volunteers of the 254 participants due to COVID-19 and hospital policies. Kindly see the CONSORT 2010 diagram in Fig 3.

Reviewer #2

Comment

1. in lines 76,77,78,79 & 80 there is repetitions that is meaningless so it needs editing to clarify your purpose.

Response: We thank you for your suggestion to review part of the introduction. Accordingly, we have revised the sentence to clearly state the purpose of this study.

Comment

2.In materials : you did not mention the number of males & females enrolled in the study.

Response: We thank you for your suggestion to review part of the materials and methods section. We included both males and females in our inclusion criteria and reported the number of each in Table 1 of the baseline characteristics.

Comment

3. In Results : you mentioned the mean age among males and females which spots the importance of sex as a variable. Moreover in table 1 the number appeared males 36 & females 91 which is of great difference

Response: We thank you for your suggestion to review part of the results section. This study aimed to determine whether the 2-minute marching and 6-minute walk tests are equivalent in assessing cardiovascular endurance. Our study did not intend to compare females and males. However, upon analyzing the results of all 127 cases, we found that both tests assess cardiovascular endurance equally. We thank you for your recommendations, and we will consider them in our next research study.

Comment

4. In Discussion : line 317 YMCA ...........?

Response: We thank you for your comment. YMCA stands for Young Men's Christian Association step test and is a standard measure of cardiovascular endurance. We have also included this abbreviation.

Comment

5. In study limitations : line 386: MCID.......?

Response: We thank you for your comment. "MCID" stands for "minimal clinically important difference." We have also included this abbreviation.

Comment

6.In references : no. 2,3,9,26,27 needs to add doi ...........

Response: We thank you for your comment. We have updated the reference style to match the journal style and included DOI and URL.

Reviewer #3: Overall, interesting study entitled A comparison of new cardiovascular endurance test using the 2-minute marching test vs. 6-minute walk test in healthy volunteers: A crossover randomized controlled trial However, the manuscript does not provide a clear overview of their work and requires revision.

Comments to authors:

Comment

1.Please make sure that the structure for citing published literature in the text, as well as the style of references in the References section, are consistent with the journal's style (see Instructions to Authors).

Response: We thank you for your valuable comment. Accordingly, we have updated the reference style to match that of the journal.

 Comment

2. English language needs revision for style and syntax

Response: The manuscript has undergone English language editing for style and syntax by a native English speaker

Comment

3.Abstract must be rewritten. Add characteristics of the participants (age…..) I suggest focusing the abstract on your study and your results

Response: We thank you for your suggestion to review a section of the abstract. We have included participant characteristics in the abstract and rewritten it based on the CONSORT 2010 abstract report for a crossover study.

Comment

4. How were the participants randomized

Response: We thank you for your important question. This was a group-randomized crossover study with implemented assignment methods.

Comment

5. Please add the originality of the study and add hypothesis at the end of the introduction section. Be please be more specific

Response: We thank you for your comment. We have revised the paragraph (Lines 95–103) to include the originality of the study and added a hypothesis as recommended by the reviewer.

Comment

6.A substantial revision of the introduction needed.

Response: We thank you for your comment. Accordingly, we have revised the introduction section.

Comment

7. Include more characteristics of participants. More information about the participant’s selection needed

Response: We thank you for your valuable comment. Ac

---

## [Decision Letter · Decision Letter 1]

31 May 2024

PONE-D-23-24966R1A comparison of new cardiovascular endurance test using the 2-minute marching test vs. 6-minute walk test in healthy volunteers: A crossover randomized controlled trialPLOS ONE

Dear Dr. Surapichpong¹¶,

Thank you for submitting your manuscript to PLOS ONE. After careful consideration, we feel that it has merit but does not fully meet PLOS ONE’s publication criteria as it currently stands. Therefore, we invite you to submit a revised version of the manuscript that addresses the points raised during the review process. Please revise comments raised by reviwer 3 Please submit your revised manuscript by Jul 15 2024 11:59PM. If you will need more time than this to complete your revisions, please reply to this message or contact the journal office at plosone@plos.org. Please include the following items when submitting your revised manuscript:A rebuttal letter that responds to each point raised by the academic editor and reviewer(s). You should upload this letter as a separate file labeled 'Response to Reviewers'.A marked-up copy of your manuscript that highlights changes made to the original version. You should upload this as a separate file labeled 'Revised Manuscript with Track Changes'.An unmarked version of your revised paper without tracked changes. You should upload this as a separate file labeled 'Manuscript'.If applicable, we recommend that you deposit your laboratory protocols in protocols.io to enhance the reproducibility of your results. Protocols.io assigns your protocol its own identifier (DOI) so that it can be cited independently in the future. For instructions see: https://journals.plos.org/plosone/s/submission-guidelines#loc-laboratory-protocols. Additionally, PLOS ONE offers an option for publishing peer-reviewed Lab Protocol articles, which describe protocols hosted on protocols.io. Read more information on sharing protocols at https://plos.org/protocols?utm_medium=editorial-email&utm_source=authorletters&utm_campaign=protocols.

We look forward to receiving your revised manuscript.

Kind regards,

Tamer I. Abo Elyazed, Ph.d

Guest Editor

PLOS ONE

Journal Requirements:

Additional Editor Comments:

Reviewers' comments:

Reviewer's Responses to Questions

**Comments to the Author**

1. If the authors have adequately addressed your comments raised in a previous round of review and you feel that this manuscript is now acceptable for publication, you may indicate that here to bypass the “Comments to the Author” section, enter your conflict of interest statement in the “Confidential to Editor” section, and submit your "Accept" recommendation.

Reviewer #1: All comments have been addressed

Reviewer #2: All comments have been addressed

Reviewer #3: All comments have been addressed

2. Is the manuscript technically sound, and do the data support the conclusions?

Reviewer #1: Partly

Reviewer #2: Yes

Reviewer #3: Yes

3. Has the statistical analysis been performed appropriately and rigorously? 

Reviewer #1: Yes

Reviewer #2: Yes

Reviewer #3: Yes

4. Have the authors made all data underlying the findings in their manuscript fully available?

Reviewer #1: Yes

Reviewer #2: Yes

Reviewer #3: Yes

5. Is the manuscript presented in an intelligible fashion and written in standard English?

Reviewer #1: Yes

Reviewer #2: Yes

Reviewer #3: Yes

6. Review Comments to the Author

Reviewer #1: (No Response)

Reviewer #2: the authors completed the comments the identification of the tests were expressed well

The results were clear

Reviewer #3: Review Comments to the Author

This version is more approved

Please check the formatting of text and tables.

7. PLOS authors have the option to publish the peer review history of their article (what does this mean?). If published, this will include your full peer review and any attached files.

Reviewer #1: No

Reviewer #2: No

Reviewer #3: No

---

## [Author Response · Author response to Decision Letter 1]

7 Jun 2024

Rebuttal letter 

Response to reviewers

We would like to express our gratitude to the editor and reviewers for taking the time to evaluate our manuscript. We have provided line-by-line responses to the comments raised. Please find our responses marked in yellow for every comment/question marked in green. We have copied the review decision from the submission menu of the editorial manager and used the same letter to write our responses. 

PONE-D-23-24966R1

A comparison of new cardiovascular endurance test using the 2-minute marching test vs. 6-minute walk test in healthy volunteers: A crossover randomized controlled trial

PLOS ONE

Dear Dr. Surapichpong¹¶,

Thank you for submitting your manuscript to PLOS ONE. After careful consideration, we feel that it has merit but does not fully meet PLOS ONE’s publication criteria as it currently stands. Therefore, we invite you to submit a revised version of the manuscript that addresses the points raised during the review process. Please revise comments raised by reviewer 3 Please submit your revised manuscript by Jul 15 2024 11:59PM. If you will need more time than this to complete your revisions, please reply to this message or contact the journal office at <a href="mailto:plosone@plos.org">plosone@plos.org. Please include the following items when submitting your revised manuscript:

We look forward to receiving your revised manuscript.

Kind regards,

Tamer I. Abo Elyazed, Ph.d

Guest Editor

PLOS ONE

Journal Requirements:

Response: Following the journal requirements, we have reviewed all references and confirmed their correctness. No references have been retracted.

Additional Editor Comments:

Reviewers' comments:

Reviewer's Responses to Questions

Comments to the Author

1. If the authors have adequately addressed your comments raised in a previous round of review and you feel that this manuscript is now acceptable for publication, you may indicate that here to bypass the “Comments to the Author” section, enter your conflict of interest statement in the “Confidential to Editor” section, and submit your "Accept" recommendation.

Reviewer #1: All comments have been addressed

Reviewer #2: All comments have been addressed

Reviewer #3: All comments have been addressed

2. Is the manuscript technically sound, and do the data support the conclusions?

Reviewer #1: Partly

Reviewer #2: Yes

Reviewer #3: Yes

3. Has the statistical analysis been performed appropriately and rigorously?

Reviewer #1: Yes

Reviewer #2: Yes

Reviewer #3: Yes

4. Have the authors made all data underlying the findings in their manuscript fully available?

Reviewer #1: Yes

Reviewer #2: Yes

Reviewer #3: Yes

5. Is the manuscript presented in an intelligible fashion and written in standard English?

Reviewer #1: Yes

Reviewer #2: Yes

Reviewer #3: Yes

6. Review Comments to the Author

Reviewer #1: (No Response)

Reviewer #2: the authors completed the comments the identification of the tests were expressed well

The results were clear

Response: Thank you for your comment. We appreciate your positive feedback, which gives us hope that this paper will be accepted for publication.

Reviewer #3: Review Comments to the Author

This version is more approved

Comment: Please check the formatting of text and tables.

Response: Thank you for your suggestion. We have adjusted the formatting of the text and tables 1-4 to comply with PLOS ONE guidelines.

7. PLOS authors have the option to publish the peer review history of their article (what does this mean?). If published, this will include your full peer review and any attached files.

Do you want your identity to be public for this peer review? For information about this choice, including consent withdrawal, please see our Privacy Policy.

Reviewer #1: No

Reviewer #2: No

Reviewer #3: No

While revising your submission, please upload your figure files to the Preflight Analysis and Conversion Engine (PACE) digital diagnostic tool, https://pacev2.apexcovantage.com/. PACE helps ensure that figures meet PLOS requirements. To use PACE, you must first register as a user. Registration is free. Then, login and navigate to the UPLOAD tab, where you will find detailed instructions on how to use the tool. If you encounter any issues or have any questions when using PACE, please email PLOS at <a href="mailto:figures@plos.org">figures@plos.org. Please note that Supporting Information files do not need this step.

Response: We have registered with PACE, and all figures are consistent with PLOS guidelines. We downloaded them from PACE and uploaded them as Fig 1, Fig 2, Fig 3, and Figures in TIFF format.

---

## [Editor Report · Decision Letter 2]

10 Jul 2024

A comparison of new cardiovascular endurance test using the 2-minute marching test vs. 6-minute walk test in healthy volunteers: A crossover randomized controlled trial

PONE-D-23-24966R2

Dear Dr. Surapichpong¹¶,

We’re pleased to inform you that your manuscript has been judged scientifically suitable for publication and will be formally accepted for publication once it meets all outstanding technical requirements.

Kind regards,

Tamer I. Abo Elyazed, Ph.d

Guest Editor

PLOS ONE
---

## [Editor Report · Acceptance letter]

16 Jul 2024

PONE-D-23-24966R2 

PLOS ONE

Dear Dr. Surapichpong, 

I'm pleased to inform you that your manuscript has been deemed suitable for publication in PLOS ONE. Congratulations! Your manuscript is now being handed over to our production team.

Kind regards, 

on behalf of

Dr. Tamer I. Abo Elyazed 

Guest Editor

PLOS ONE